# The Valorization of Potato Peels as a Functional Ingredient in the Food Industry: A Comprehensive Review

**DOI:** 10.3390/foods14081333

**Published:** 2025-04-12

**Authors:** Domizia Vescovo, Cesare Manetti, Roberto Ruggieri, Umile Gianfranco Spizzirri, Francesca Aiello, Maria Martuscelli, Donatella Restuccia

**Affiliations:** 1Department of Management, Sapienza University of Rome, 00161 Rome, Italy; domizia.vescovo@uniroma1.it (D.V.); roberto.ruggieri@uniroma1.it (R.R.); 2Department of Environmental Biology, Sapienza University of Rome, 00161 Rome, Italy; cesare.manetti@uniroma1.it; 3Ionian Department of Law, Economics and Environment, University of Bari Aldo Moro, 74123 Taranto, Italy; umile.spizzirri@uniba.it; 4Department of Pharmacy, Health and Nutritional Sciences, University of Calabria, 87036 Rende, Italy; francesca.aiello@unical.it; 5Department of Bioscience and Technology for Food, Agriculture and Environment, University of Teramo, 64100 Teramo, Italy; mmartuscelli@unite.it

**Keywords:** agri-food waste, potato peels, extraction, bioactive compounds, functional food, circular economy

## Abstract

Potato peels (PPs) represent a significant agro-industrial by-product with notable potential for valorization due to their rich composition of bioactive compounds, including phenolics, glycoalkaloids, dietary fiber, and essential minerals. This review explores the functional applications of PPs in the food industry by examining their chemical profile, extraction methods, and biological activities. Phenolic compounds, mainly chlorogenic acid and its derivatives, are the most abundant bioactives and contribute to antioxidant and anti-inflammatory properties. Glycoalkaloids, such as *α*-solanine and *α*-chaconine, exhibit antimicrobial activity but require careful monitoring due to their potential toxicity, although recent evidence suggests that controlled doses may provide health benefits. The choice of extraction technique influences the recovery of these compounds, with ultrasound-assisted extraction (UAE) and microwave-assisted extraction (MAE) proving to be efficient and environmentally friendly alternatives to conventional methods. The incorporation of PP-derived ingredients into food formulations, including cereal, dairy, meat, and fish products, as well as vegetable oils, has shown promising results in the improvement of nutritional quality, oxidative stability and functional properties. However, challenges remain with regard to the standardization of PP composition, bioavailability of bioactive compounds and their stability within food matrices. Advancing research on PPs will not only contribute to circular economy goals but also provide innovative solutions for the food industry, reinforcing the link between sustainability and human health.

## 1. Introduction

By 2050, the demand for food production is expected to increase by approximately 55% compared to 2010, due to population growth, economic development, and urbanization [1]. This will lead to negative impacts on both the environment and biodiversity. To prevent irreversible damage, the rise in food production should be coupled with targeted investment policies promoting sustainable and circular production practices [1]. In this regard, minimizing food loss and waste is widely considered a valuable strategy to reduce production costs, enhance the efficiency of the food system, improve nutrition and food security, and contribute to environmental sustainability [2,3]. At the global level, the UN Environment Programme’s Food Waste Index Report showed that around 1.052 billion tons of food was wasted in 2022, equivalent to 132 kg per capita [4]. It has been estimated that almost a third of the annual production for human consumption is lost or wasted every year [5]. The growing attention to food loss and waste is reflected in the Sustainable Development Goals (SDGs) defined in the UN 2030 Agenda. Sustainable Development Goal 12.3 aims to halve global food waste per capita at the retail and consumption levels and reduce food losses along production and supply chains (including post-harvest losses) by 2030. Reducing food loss and waste can also contribute to the achievement of other SDGs, including SDG 2 “Zero hunger”, SDG 6 “Sustainable water management”, SDG 13 “Climate change”, and SDG 15 “Terrestrial ecosystems, forestry, biodiversity” [5].

However, other kinds of agro-industrial or processing residues should be considered event though they are not regarded as food loss or waste. The total volume of agricultural residues, defined as parts of crops not intended for human consumption, reaches 367 million tons annually within Europe. A significant proportion of these residues is repurposed directly on farms, commonly as livestock bedding, animal fodder, or for various horticultural applications [6].

More recently, the scientific and industrial community increasingly focused on the recovery of bioactive compounds present in agro-industrial and processing wastes, such as leaves, peels, seeds, pulp, and skin or shell, to reduce their environmental impact [7,8,9]. These compounds can be usefully employed as ingredients in functional foods or nutraceuticals [10,11,12,13]. These solutions align with the concept of a circular bioeconomy, aiming to reduce environmental, social, and economic costs, enhance economic competitiveness, and alleviate poverty and hunger. This is in line with the “*Waste-to-Wealth*” approach, which attempts to extract valuable resources from waste [14,15,16]. Thus, the valorization of agro-industrial by-products is one of the main objectives of the food industry, which focuses considerable effort on developing healthier foods using bioactive compounds.

The potato industry is one of those which generates a significant amount of waste that is potentially harmful to the environment. Potatoes (*Solanum tuberosum* L.) are considered the fourth most important crop worldwide, after rice, wheat, and maize. According to FAO data, 383 million tons of potatoes were produced globally in 2023 [17]. The world’s leading producer is China, with an annual production of approximately 93.5 million tons [17]. China allocates a large portion of its potato production to industrial processing for products such as starch, chips, and french fries. Other major producers include India, with 60.1 million tons, and the United States, with 19.9 million tons, where a significant share of production is destined for processing plants [17]. The European Union is also a significant production area, with the potato industry valued at around EUR 9 billion in 2019, a figure that represents approximately 1.6% of the total value of the European food industry [18]. Around 50% of the world’s potato production is processed into products such as chips, french fries, starches, and purée. Potatoes can grow in different climatic and growing conditions, and they have high nutritional value. The potato flesh (PF) is an excellent source of carbohydrates, proteins, vitamins, minerals, and dietary fiber, as well as antioxidants [19].

Potato processing activities generate significant amounts of peels, ranging from 15% to 40% (*w*/*w*) of the original fresh weight (FW), depending on the peeling process [20]. In particular, it is estimated that by 2030 approximately 8000 kilo tons of potato peels (PPs) could be generated, with associated greenhouse gas emissions of 5 million tons of CO_2_eq [21]. Most of these wastes are disposed in landfills or used as animal feed, with low added value in the production chain. More recently, the potential use of potato processing residues as biomass for energy and biogas production has been explored [22].

As illustrated in Figure 1**,** PPs are not merely a by-product, but a valuable source of nutrients and bioactive compounds. They contain significant amounts of starch, dietary fiber (mainly insoluble), high-biological-value proteins, and minerals. Notably, they are particularly rich in phenolic compounds, with levels up to ten times higher than those found in PF, and glycoalkaloids, which have been associated with various biological activities [22,23,24].

Considering the growing interest in research on agricultural by-products and their potential benefits, this review aims to comprehensively assess the current state of knowledge on the valorization of potato peels (PPs) as additives in functional food. Specifically, it will analyze their nutritional and bioactive composition, potential applications in the food industry, and their role in promoting sustainability through waste reduction and circular bioeconomy strategies. This review will also discuss the main challenges and future perspectives related to the utilization of PP-derived bioactive compounds in food product development. In light of global challenges such as food security, climate change, and resource efficiency, these findings highlight the importance of valorizing agro-industrial by-products for both environmental and health-related benefits.

## 2. Potato Chemical Composition

Potatoes are a good dietary source of carbohydrates, which constitute approximately 75% of the total dry matter of the tuber. Starch is the predominant carbohydrate in potatoes, with a content ranging from 16.5 to 20.0 g/100 g of FW [25]. The protein fraction varies from 1% to 1.5% FW of the tuber [26]. Despite the relatively low amount, potato proteins have an excellent biological value (BV) of 90–100, similar to that of whole egg (100), and higher than that of soybeans (84) and legumes (74) [27]. Total lipids are low and range from 0.1 to 0.5 g/100 g FW [28], with the prevalence of polyunsaturated fatty acids n-6 and n-3 [25]. Dietary fiber is primarily provided by the thickened cell walls of the peel, which constitute 1–2% of the tuber. The fiber content can range from 1.8 to 3.30 g/100 g FW [25,29,30]. These data are consistent with the values reported in the USDA and CREA databases (Table 1).

The primary minerals found in raw potatoes include potassium (150 to 186 mg/100 g FW), phosphorus (42 to 120 mg/100 g FW), magnesium (16 to 40 mg/100 g FW), and calcium (2 to 20 mg/100 g FW) [25]. Fresh potatoes have variable concentrations of vitamin C, which can reach nearly 50 mg/100 g FW [31]. Vitamin B_6_ is also significantly present in potatoes, with concentrations ranging from 0.450 to 0.675 mg/100 g FW [32]. Additionally, potatoes are rich in bioactive compounds such as flavonoids, carotenoids, and phenolic acids. These compounds contribute to their antioxidant properties and potential health benefits, which have been well documented in scientific studies [33,34]. In particular, lutein and zeaxanthin are present at high levels in yellow-fleshed potatoes, and anthocyanins are found in purple- and red-fleshed varieties. Potatoes also contain glycoalkaloids, which can be toxic to humans when present in high amounts. However, in smaller quantities, these compounds have been shown to offer potential health benefits, including the ability to inhibit the growth of cancer cells [35]. Like other plant foods, the nutritional composition of potatoes is influenced by various pre-harvest conditions (genotype, environment, cultivation practices, maturity at harvest, biotic and abiotic stress, etc.) and post-harvest factors (processing, storage, transport, etc.) [25,36].

As far as PPs are concerned, their chemical composition is reported in Table 1. As can be observed, they contain significant amounts of starch, dietary fiber, proteins, and bioactive compounds. It follows that they can be incorporated into the diet through food functionalization [22]. Among these components, carbohydrates are identified as the most abundant macronutrients in PP. Analysis of the available literature data shows that the carbohydrate content, considering varietal differences, falls within similar ranges: 63.0 g/100 g dry weight (DW) [37], 68.7 g/100 g DW [23], 72.0 g/100 g DW [36], 69.0–88.0 g/100 g DW [22], and 46.2–77.4 g/100 g [18]. Low-molecular-weight free sugars, or reducing sugars, constitute a minor proportion of the total carbohydrates, with maltose, sucrose, glucose, and fructose being the predominant reducing sugars [23]. Starch accounts for approximately 50% (*w*/*w*) of the DW of the total carbohydrate content [38,39].

PPs are a good source of dietary fiber, both soluble—such as pectins and gums (important for cholesterol reduction and glucose regulation)—and insoluble—such as cellulose, hemicellulose, and lignin (regulators of the intestinal tract) [23,40,41]. The quantification of total dietary fiber varies considerably among different varieties and different studies. Some studies report an average content of approximately 2.5–3.3 g/100 g DW [38,41]. However, as shown in Table 1, the values can vary significantly. This variability depends not only on methodological differences but also on which compounds researchers classify as ‘total dietary fiber’. Similarly, the protein content in PP shows considerable variability across studies, ranging from 2 to 17 g/100 g DW [18,22,42].

The fat content is very low, ranging from 0.8 to 2.9 g/100 g DW [18,43,44]. Ash content is an indicator of mineral composition, with values reported between 4.83% and 11.05% DW [18,43]. In particular, the thick periderm of PP has been found to contain minerals such as calcium (1% *w*/*w*), iron (6%), magnesium (6%), manganese (7%), phosphorus (8%), potassium (9%), and zinc (3%), as well as several vitamins: B_1_, B_2_, B_3_, B_5_, B_6_, C, K, and folate (B_9_) [21,23].

**Table 1 foods-14-01333-t001:** Chemical composition of potatoes (flesh, peels, flesh/peels).

Potato Cultivar		Carbohydrates	Protein	Fiber	Fat	Ash	Unit	References
**Flesh**								
*Gold Potato* (*Solanum tuberosum* L.) raw, without skin		16.0	1.81	13.8	0.26	0.89	g/100 g FW	[45]
*Red Potato* (*Solanum tuberosum* L.) raw, without skin		16.3	2.06	13.8	0.25	0.95	g/100 g FW	[45]
*Primura* (*Solanum tuberosum* L.) raw		16.0	2.0	1.8	0.1	n.r.	g/100 g FW	[46]
*Marabel* (*Solanum tuberosum* L.) raw		16.0	2.0	1.8	0.1	n.r.	g/100 g FW	[46]
*Agata* (*Solanum tuberosum* L.) raw		16.0	2.0	1.8	0.1	n.r.	g/100 g FW	[46]
*Annabelle* (*Solanum tuberosum* L.) raw		16.0	2.0	1.8	0.1	n.r.	g/100 g FW	[46]
*Vivaldi* (*Solanum tuberosum* L.) raw		16.0	2.0	1.8	0.1	n.r.	g/100 g FW	[46]
**Peels**								
*Red Potato* (*Solanum tuberosum* L.)		72 *	15.99 *	15.97 *	0.81 *	6.69 *	%	[47]
*Gold Potato* (*Solanum tuberosum* L.)		70 *	14.17 *	21.72 *	1.17 *	9.12 *	%	[47]
*Organic Russet* (*Solanum tuberosum* L.)		76 *	11.98 *	21.4 *	1.12 *	7.32 *	%	[47]
*Non-organic Russet* (*Solanum tuberosum* L.)		71 *	17.19 *	22.39 *	1.1 *	7.34 *	%	[47]
*Lady Rosetta* (*Solanum tuberosum* L.)		72.53 ± 0.08	11.17 ± 0.03	n.r.	2.09 ± 0.01	7.24 ± 0.02	g/100 g DW	[48]
*Lady Claire* (*Solanum tuberosum* L.)		77.38 ± 0.65	12.44 ± 0.09	n.r.	1.27 ± 0.38	4.83 ± 0.13	g/100 g DW	[48]
*Spunta* (*Solanum tuberosum* L.)		88.0 ± 4.4	2.10 ± 0.11	n.r.	0.73 ± 0.04	0.91 ± 0.01	% DW	[49]
*Agria* (*Solanum tuberosum* L.)		86.97 ± 0.43	6.47 ± 0.23	n.r.	0 *	5.46 ± 0.17	% DW	[50]
U.V. (*Solanum tuberosum* L.)		68.7 *	8 *	n.r.	2.6 *	6.34 *	% DW	[39]
U.V. (*Solanum tuberosum* L.)		73.79 ± 3.02	4.42 ± 0.45	4.81 ± 0.07	1.63 ± 0.54	3.65 ± 0.41	%	[51]
U.V. (*Solanum tuberosum* L.)		43.20 *	10.73 *	n.r.	2.45 *	7.45 *	%	[52]
*Purple-Fleshed Sweet Potato* (*Ipomoea batatas* L.)		79.10 ± 0.03	2.33 ± 0.04	15.8 ± 0.50	0.51 ± 0.04	2.06 ± 0.01	g/100 g DW	[36]
*Sweet Potato* (*Ipomoea batatas* L.)		72.60 ± 1.58	4.64 ± 0.51	3.79 ± 0.67	2.02 ± 0.22	4.56 ± 1.15	%	[51]
**Flesh/Peels**	**Potato tissue**							
*Red Emmalie* (*Solanum tuberosum* L.)	Flesh	5.03 TSS °Brix	103.3 *	24.2 *	n.r.	61.4 *	g/kg DM	[24]
	Peels	3.53 TSS °Brix	147.2 *	66.4 *	n.r.	78.2 *	g/kg DM	[24]
*Rosalinde* (*Solanum tuberosum* L.)	Flesh	4.50 TSS °Brix	121.1 *	21.4 *	n.r.	52.1 *	g/kg DM	[24]
	Peels	4.16 TSS °Brix	161.1 *	71.7 *	n.r.	76.6 *	g/kg DM	[24]
*Highland Burgundy Red* (*Solanum tuberosum* L.)	Flesh	5.40 TSS °Brix	107.1 *	27.6 *	n.r.	55.7 *	g/kg DM	[24]
	Peels	4.56 TSS °Brix	165.3 *	75.8 *	n.r.	86.5 *	g/kg DM	[24]
*Violetta* (*Solanum tuberosum* L.)	Flesh	4.86 TSS °Brix	119.3 *	26.5 *	n.r.	67.0 *	g/kg DM	[24]
	Peels	3.63 TSS °Brix	148.9 *	73.7 *	n.r.	92.5 *	g/kg DM	[24]
*Valfi* (*Solanum tuberosum* L.)	Flesh	4.73 TSS °Brix	89.4 *	20.3 *	n.r.	51.5 *	g/kg DM	[24]
	Peels	4.46 TSS °Brix	139.0 *	55.5 *	n.r.	68.2 *	g/kg DM	[24]
*Salad Blue* (*Solanum tuberosum* L.)	Flesh	5.06 TSS °Brix	119.4*	20.5*	n.r.	52.5 *	g/kg DM	[24]
	Peels	3.86 TSS °Brix	141.1 *	47.5 *	n.r.	6.33 *	g/kg DM	[24]
*Maris Piper* (*Solanum tuberosum* L.)	Flesh	76.4 ± 3.3	7.1 ± 0.21	n.r.	2.61 ± 0.05	n.r.	% (*w*/*w*) DB	[44]
	Peels	62.4 ± 3.3	11.0 ± 0.15	n.r.	1.75 ± 0.02	n.r.	% (*w*/*w*) DB	[44]

* Standard deviation not reported; U.V.: unknown variety; n.r.: not reported; FW: fresh weight; DW: dry weight; DM: dry matter; DB: dry basis.

### 2.1. Bioactive Compounds of PP

As already stated, PPs are rich in bioactive molecules, mainly phenolic compounds and glycoalkaloids.

Phenolic compounds are a broad class of plant secondary metabolites, characterized by the presence of an aromatic ring with one or more hydroxyl groups. These compounds are divided into non-flavonoids, such as hydroxycinnamic acids (e.g., chlorogenic, caffeic, ferulic, and *p*-coumaric acids) and hydroxybenzoic acids (e.g., gallic, vanillic, protocatechuic, and *p*-hydroxybenzoic acids), and flavonoids, i.e., flavonols, flavanols, flavones, isoflavones, and anthocyanins [53]. Their primary role is to defend plants against environmental stresses, such as pathogen infections, herbivores, and nutrient deficiencies [54]. In fact, phenolic compounds demonstrate antimicrobial and antioxidant properties, enabling plants to resist pathogenic infections and safeguard their primary tissues against the harmful effects of reactive oxygen species [55]. These molecules have attracted considerable attention for their health-promoting effects in a range of chronic conditions, including diabetes, cardiovascular diseases, cancer, and neurodegenerative disorders, among others. These benefits are largely attributed to their antioxidant properties. Phenolic compounds act as antioxidants by neutralizing free radicals, chelating metal ions, and enhancing the activity of the endogenous antioxidant system, which includes both enzymatic antioxidants (such as catalase, superoxide dismutase, and glutathione peroxidase) and non-enzymatic antioxidants (such as glutathione) [56].

Glycoalkaloids are another group of plant-derived secondary metabolites, consisting of a steroidal alkaloid structure conjugated to one or more sugar molecules through glycosidic bonds. These compounds are naturally synthesized in tubers during the germination phase, serving a primary role in the plant’s defense mechanisms against bacterial, fungal, and insect attacks [57]. However, glycoalkaloids exhibit a dual nature, with both toxic and beneficial effects. Toxicity is linked to mechanisms such as acetylcholinesterase inhibition and cell membrane disruption, which can negatively impact the nervous and digestive systems. At high concentrations (above 200 mg kg^−1^ FW), glycoalkaloids can lead to neurological symptoms such as drowsiness, apathy, and disorientation, with severe cases potentially being fatal. At the same time, glycoalkaloids also demonstrate promising bioactivities, including anticarcinogenic effects mediated through similar mechanisms [21,38]. In recent years, numerous in vitro and preclinical studies have explored the health-promoting properties of these compounds. They have shown antibacterial, antifungal, anti-inflammatory, and anticancer activities, with significant potential in addressing metabolic disorders, microbial infections, glycemia and allergies. Among glycoalkaloids, α-chaconine has emerged as particularly potent, exhibiting up to five times the bioactivity of α-solanine [22,58]. To ensure consumer safety, monitoring glycoalkaloid concentrations in potato products is crucial. Recommended limits for glycoalkaloids are set at 100 mg kg^−1^ FW for standard consumption, with an upper safety threshold of 200 mg kg^−1^ FW to minimize toxicological risks [59]. A more recent recommendation from the European Food Safety Authority [60] states that the acceptable daily intake of total glycoalkaloids should not exceed 1 mg kg^−1^ of body weight in humans, a concentration at which no adverse effects have been observed. These regulations aim to balance safety with the potential therapeutic applications of glycoalkaloids. However, further research, including in vivo studies and applications in food systems at varying concentrations, is needed to evaluate their feasibility for industrial use [60].

#### 2.1.1. Phenolic Compounds in PP

As reported in the literature, the concentrations of phenolic compounds in PP vary widely. Discrepancies can be mainly related to intrinsic factors, such as potato variety and degree of maturation. For instance, PPs derived from red- and purple-fleshed varieties accumulate anthocyanins and contain nearly twice the concentration of phenolic compounds compared to PPs of yellow-fleshed varieties [21]. At the same time, peels of young potatoes presented twice the phenolic content of mature potatoes [61]. Chlorogenic acid content can also increase during early maturation and decrease with over-ripening [62].

Additionally, extrinsic factors such as the extraction methodologies and the specific compounds targeted for the analysis further contribute to the observed variability [63].

PP are reported to contain phenolic compounds at concentrations up to 10 times higher than those found in the flesh [43]. This has led to an increasing interest in PP as a rich source of natural antioxidants, with potential applications as functional ingredients in food formulations [38]. The predominant phenolic class in PP is represented by phenolic acids [64,65,66]. Chlorogenic acid, which is the ester of caffeic acid and quinic acid, has been extensively described in potatoes [21,22,50,67,68]. It constitutes 90% of the phenolic compounds in PP and exists in the form of three main isomers: chlorogenic acid (5-*O*-caffeoylquinic acid), neochlorogenic acid (3-*O*-caffeoylquinic acid), and cryptochlorogenic acid (4-*O*-caffeoylquinic acid) [66,69,70]. In contrast, other researchers, such as Riciputi et al. [71], have reported that chlorogenic acid constitutes 49.3–61% of the total phenolic compounds. These discrepancies may be attributed to the partial degradation of chlorogenic acid into caffeic acid under specific conditions, such as exposure to direct light or high temperatures [22]. In addition to chlorogenic and caffeic acids, PP contain other significant phenolic acids, including gallic, *p*-coumaric, ferulic, protocatechuic, and vanillic acids. The quantification of total phenolics is highly variable in the literature (Table 2): 0.687–14.0 mg_GAE_ g^−1^ DB [18], 3.2–10.3 mg_GAE_/100 g DB (*Agria* variety) [50], 48 mg/100 g DM [23], 1.976 mg/100 g of DW [65], and 29.24 mg_GAE_ per gram of extract [21].

Flavonoids, including flavonols, flavanols, and anthocyanins, have also been identified in potatoes, although their concentrations are generally lower than those of phenolic and chlorogenic acids. Flavonols, such as quercetin-3-rutinoside (rutin), quercetin, myricetin, and kaempferol-3-rutinoside, have been reported in specific varieties at levels ranging from 0.6 to 21 mg/100 g DW [70]. Silva-Beltràn et al. [72], reported a total flavonoid content expressed as quercetin equivalents (QE) at 3.3 mg QE g^−1^, whereas other authors expressed flavonoid content in catechin equivalents (CE). For instance, Khanal et al. [21] obtained total flavonoid levels in PP ranging from 0.06 to 2.29 mg CE/g DW. Anthocyanin content in colored-flesh potatoes can vary widely as well, with values reaching up to 1600 mg/100 g DW or ranging from 20 to 400 mg kg^−1^ FW and 55 to 350 mg kg^−1^ FW, depending on the variety [62,70].

**Table 2 foods-14-01333-t002:** The contents of phenolic compounds in PP as reported in the literature.

PP Variety and Origin	Chlorogenic Acid	Caffeic Acid	Ferulic Acid	Gallic Acid	Protocatechuic Acid	Vanillic Acid	p-Hydroxybenzoic Acid	p-Coumaric Acid	Catechin	Quercetin	Rutin	Reference
*Bintje*— Italy (mg/g DW)	1.97 ± 0.02	0.24 ± 0.00	0.06 ± 0.00	-	-	-	-	-	-	-	-	[71]
*Challenger*— Italy (mg/g DW)	1.27 ± 0.01	0.22 ± 0.00	0.05 ± 0.00	-	-	-	-	-	-	-	-	[71]
*Daisy*— Italy (mg/g DW)	4.10 ± 0.03	0.16 ± 0.00	0.12 ± 0.00	-	-	-	-	-	-	-	-	[71]
*Innovator*— Italy (mg/g DW)	2.52 ± 0.01	0.30 ± 0.05	0.06 ± 0.00	-	-	-	-	-	-	-	-	[71]
*Fontane*— Italy (mg/g DW)	3.04 ± 0.01	1.22 ± 0.01	0.12 ± 0.00	-	-	-	-	-	-	-	-	[71]
*Maris Piper*— UK (mg/g DW)	3.87 *	0.92 *	-	-	-	0.31 *	-	traces	-	-	-	[44]
*Spunta*—Greece (mg/100 g DW)	8.3 ± 0.5	8.9 ± 0.2	10.8 ± 3.5	-	-	-	-	-	-	-	-	[73]
*Spunta* —Luxembourg (µg/g DW)	907 ± 200	104 ± 41.9	2.5 ± 2.0	-	-	1.5 ± 1.4	-	4.7 ± 2.0	3.8 ± 5.3	-	11.2 ± 6.6	[74]
*Bintje*—Luxembourg (µg/g DW)	948 ± 169	103± 21.0	0.6 ± 0.2	-	-	3.1 ± 0.7	-	1.4 ± 0.4	1.8 ± 1.8	-	5.3 ± 2.1	[74]
*Russet*— Canada (mg/100 g freeze-dried sample)	134.9 *	98.5 * (a)	56.9 * (a)	-	-	-	-	8.7 * (a)	-	-	-	[75]
*Innovator*— Canada (mg/100 g freeze-dried sample)	128.9 *	109.4 * (a)	84.8 * (a)	-	-	-	-	5.3 * (a)	-	-	-	[75]
*Yellow*— Canada (mg/100 g freeze-dried sample)	16.9 * (a)	29.7 * (a)	12.5 * (a)	-	-	-	-	2.6 * (a)	-	-	-	[75]
*Purple*— Canada (mg/100 g freeze-dried sample)	364.9 * (a)	92 *(a)	6.9 * (a)	-	-	-	-	7.4 * (a)	-	-	-	[75]
*Fianna*— Mexico (mg/100 g dw acidified ethanol extracts)	346.03 ± 2.14	332.58 ± 3.67	3.29 ± 0.05	233.49 ± 9.78	-	-	-	-	-	11.22 ± 0.09	5.01 ± 1.03	[72]
*Fianna*— Mexico (mg/100 g DW water extracts)	159.99 ± 1.05	56.99 ± 3.23	-	39.99 ± 3.03	-	-	-	-	-	2.18 ± 0.07	-	[72]
*Agria* (mg/L ethanolic extracts)	0.56–31.94	-	0.36–6.41	-	-	-	-	-	-	-	-	[64]
*Kennebec*—Luxembourg (µg/g DW)	1625 ± 323	233 ± 19.5	1.9 ± 1.3	-	-	1.5 ± 1.9	-	3.0 ± 0.8	1.9 ± 2.7	-	120.0 ± 4.4	[74]
*Mona Lisa*—Luxembourg (µg/g DW)	611 ± 149	140 ± 21.8	3.2 ± 3.6	-	-	4.6 ± 0.9	-	0.6 ± 0.7	1.2 ± 1.7	-	31.9 ± 3.2	[74]
U.V.—Spain (μg/mg dry extract)	6.013 ± 0.189	1.297 ± 0.086	0.152 ± 0.010	-	-	0.067 ± 0.005	-	0.031 ± 0.002	-	-	-	[76]
Average of different varieties (mg/100 g DB)	1.0–620	3.3–333	0.02–3.3	40–233	~2 mg/100 g fw *	0.12–31	-	0.02–3.3	-	11 *	12 *	[18]
Average of different varieties (mg/100 g)	100.0–220.0	5.0–50	0.6–9	-	1.3–7.6	1.6–22.4	-	-	-	-	-	[43]
Average of different varieties (mg/100 g DW)	1468.1 ± 39.3	172.4 ± 3.2	3.9 ± 2.5	-	7.6 ± 0.5	22.4 ± 2.5	7.84 ± 1.27	1.6 ± 0.9	-	-	-	[65]
Average of different varieties (mg/100 g DW)	753.0–821.3	278.0–296.0	174.0–192.0	58.6–63.0	216.0–256.0	43.0–48.0	82.0–87.0	41.8–45.6	-	-	-	[41,43]

* Standard deviation not reported. (-): Not reported or below the limit of quantification; (a): the value shows the sum of the free, esterified, and bound forms; U.V.: unknown variety; FW: fresh weight; DW: dry weight; DB: dry basis.

#### 2.1.2. Glycoalkaloids in PP

As observed for phenolic compounds, the glycoalkaloid concentration in both potato flesh and peels shows considerable variation due to extrinsic factors, including light exposure, storage conditions, mechanical damage, and irradiation [77]. The most significant glycoalkaloids found in PP are *α*-solanine and *α*-chaconine, together accounting for approximately 90% of the total glycoalkaloid content (TGA). Deußer et al. [74] reported significant variation in TGAs among different samples, underscoring the impact of variety on the formation of these bioactive compounds. They also demonstrated that PPs contain 3.9 to 48.7 times more glycoalkaloids than the outer flesh. To this regard, Salem et al. [23] demonstrated a heterogeneous distribution of these compounds with significantly higher concentrations in the peel compared to the whole tuber and flesh, suggesting a preferential accumulation in the outer potato layers. The total concentration of glycoalkaloids in PPs ranged from 40.9 to 5342 mg kg^−1^ DB as reported by Almeida et al. [18]. In a review by Jimenez-Champi et al. [73], TGAs, expressed as the sum of *α*-chaconine and *α*-solanine, was recorded for various PPs of different potato varieties, and the values ranged from 6.71 to 3580 μg g^−1^ DW. Specifically, *α*-solanine and α-chaconine can be found in PP at concentrations up to 3229 mg kg^−1^ DW for *α*-solanine and 4014 mg kg^−1^ DW for *α*-chaconine [18]. In a similar study, Friedman et al. [78] reported α-chaconine concentrations in PP ranging from 424 to 1297 μg g^−1^ DW in conventional potatoes and from 610 to 2830 μg g^−1^ DW in organic potatoes. The concentration of α-solanine was reported to range from 215 to 412 μg g^−1^ DW in conventional peels and from 239 to 750 μg/g DW in organic peels. Martínez-García et al. [79] reported that *α*-solanine levels in PP ranged from 143 to 12.73 mg kg^−1^ DW, while *α*-chaconine levels varied from 117 to 1742 mg kg^−1^ DW. No significant differences were observed based on peel color (yellow or red) or potato size (large or small). In Table 3, the main findings reported in the literature are collected.

## 3. Extraction of Bioactive Compounds

The extraction procedure represents a crucial step for the recovery of the bioactive substances from the PP matrix, also accounting for the significant variability previously underlined. In particular, the choice of solvent, the extraction technique, and the operating conditions (e.g., temperature, duration, solvent concentration, and matrix/solvent ratio) emerged as crucial parameters impacting both the yield and composition of the extracts [73,82,83]. Figure 2 illustrates a typical extraction process, highlighting the key steps involved in the extraction of phenolic compounds.

### 3.1. Conventional Extraction Methods

Irrespective of the kind of sample, the extraction methods for bioactive compound recovery can be classified into conventional and non-conventional techniques. Conventional methods are based on solid–liquid extraction (SLE), involving the reflux of large amounts of solvents. In contrast, non-conventional methods disrupt cellular structures, facilitating the release of intracellular compounds and enabling a more efficient and selective extraction process [64]. SLE is characterized by high solvent consumption and prolonged extraction times, which represent its main drawbacks. Despite these limitations, SLE remains widely employed due to its simplicity, cost-effectiveness, and low initial investment requirements. Phenolic compounds, due to their high content of hydroxyl groups, are usefully extracted by polar solvents, like methanol, ethanol, alcoholic mixtures, and acetone [38,66,73,83]. In the study by Mohdaly et al. [84], the highest yield of phenolic compounds, expressed as total phenolic content (TPC), from PP was obtained using methanol, followed by ethanol, acetone, diethyl ether, petroleum ether, and hexane. Similarly, in the study by Kumari et al. [48], four solvent combinations were tested in SLE: 100% distilled water, 100% methanol, 80% methanol/water, and 50% methanol/water (*v*/*v*). Among these, the 80% methanol/water combination was identified as the most effective for extracting phenolic compounds from PP. In agreement with these findings, Joly et al. [85] reported that the greatest amounts of chlorogenic acids were obtained when PPs were treated with methanol, further confirming the effectiveness of this solvent in phenolic compound recovery. Other studies, however, suggest that ethanol may be a more effective solvent. For instance, Silva-Beltrán et al. [72] found that acidified ethanol provided higher extraction yields of phenolic compounds from PP compared to water. Similarly, Brahmi et al. [82] identified that the 80% (*v*/*v*) ethanol/water mixture was the best concentration for extracting phenolic compounds from PP. Further details and a comprehensive overview of the studies can be found in Table 4.

In addition to phenolic compounds, glycoalkaloids can also be extracted from PP using SLE. Apel et al. [91] investigated and optimized SLE as a method to recover glycoalkaloids from PP. They determined that the optimal conditions for glycoalkaloid recovery during SLE were an extraction time of 76.6 min and a temperature of 83.5 °C. The study further emphasizes that increasing the extraction temperature enhances the recovery of glycoalkaloids, and longer extraction times at higher temperatures further improved the process. These findings align with previous research on glycoalkaloid extraction and other secondary metabolites, suggesting that elevated temperatures facilitate the extraction by disrupting cell walls and increasing diffusion rates [91]. In a study by Friedman et al. [92], PP samples contained TGA ranging from 1.2 to 5.3 mg g^−1^ on a dry weight basis, with α-chaconine present at approximately twice the concentration of α-solanine. Given the widely recognized influence of genetic factors, as well as pre-harvest and post-harvest conditions on glycoalkaloid content in PP, such significant variations in glycoalkaloid levels were to be expected [92]. Table 5 summarizes the operating conditions and results from various studies reported in the literature.

### 3.2. Non-Conventional Extraction Methods

To overcome the limitations of SLE, several non-conventional extraction techniques were developed and applied to isolate bioactive compounds from PP. Among the most promising methods, ultrasound-assisted extraction (UAE), microwave-assisted extraction (MAE), pressurized liquid extraction (PLE) and supercritical fluid extraction (SFE) play a crucial role. These modern approaches were particularly valuable in the valorization of by-products, as they significantly reduce the use of organic solvents, thereby improving extraction efficiency and minimizing environmental and health-related risks associated with the solvent toxicity [38].

UAE is a simple and efficient green technology that can replace traditional shaking or heating steps. The application of ultrasound greater facilitates solvent penetration into the sample matrix, increasing the contact surface area between the solid and liquid phases. This process accelerates the diffusion of solutes from the solid matrix into the solvent, significantly enhancing extraction yields. Moreover, UAE enables processing at lower temperatures, offering a key advantage in preserving heat-sensitive bioactive compounds and food components [66]. Extraction temperature, duration, and amplitude are key parameters in UAE that influence the recovery efficiency of secondary metabolites from plant tissues [91]. UAE substantially enhances the recovery of bioactive compounds from PP when compared to traditional extraction methods [38]. This is evident when comparing the results from Kumari et al. [48], where the TPC in SLE extracts for the *Lady Rosetta* variety was 3.28 mg_GAE_ g^−1^ DB, while UAE showed a TPC which increased to 7.67 mg_GAE_ g^−1^ DB (Table 6). For the *Lady Claire* variety, the TPC increased from 2.17 to 4.24 mg_GAE_ g^−1^ DB after undergoing UAE.

Regarding glycoalkaloids, the study by Apel et al. [91] revealed that yields of α-solanine, α-chaconine, and TGA from optimized SLE were significantly higher (35.2%, 63.2%, and 45.8%, respectively) compared to those from optimized UAE. These contrasts are mainly due to the experimental conditions that involve higher temperatures and longer extraction times for SLE. For instance, Martínez-García et al. [79] suggested that optimal UAE conditions involved methanol (MeOH) as a solvent, a sample-to-solvent ratio of 1:10 (*w*/*v*), and an extraction time equal to 2.5 min. These findings highlighted the importance of the extraction parameters for glycoalkaloid recovery and suggested that if SLE offers cost-effective, industrial-scale solutions, UAE methodology requires further optimization to achieve uniform cavitation and scalable efficiency. From an economic and environmental perspective, UAE is considered a viable alternative due to its low solvent consumption and reduced extraction time. However, the high initial equipment costs may limit its large-scale adoption [93].

MAE employs microwave energy, a form of non-ionizing radiation, to penetrate materials without changing the chemical structure of their essential components. The interaction of the microwaves with the molecules, through ionic conduction and dipole rotation, results in a temperature increase. This rise in temperature, along with the pressure generated by water vapor against cell walls, promotes the release of bioactive compounds into the solvent [83]. Although MAE has been extensively documented for extracting bioactive compounds from various agro-industrial wastes, its use in processing by-products from the potato industry remains limited [83]. Nonetheless, Singh et al. [94] demonstrated that MAE effectively extracts selected phenolics from PP, reducing both solvent usage and extraction time, while enhancing selectivity for target molecules. In their study, the highest TPC (3.94 mg_GAE_ g^−1^ DW) was achieved using a 67.33% (*v*/*v*) methanol solution at a microwave power level of 14.67% for 15 min. For glycoalkaloids, Guan et al. [95] conducted a combined UAE and MAE study, employing a solvent mixture of acetonitrile/formic acid (99:1) and water in a 5:1 ratio, achieving a TGA content of 292.91 mg kg^−1^ DB, slightly below the values recorded using UAE alone. MAE provides an eco-friendly approach by reducing solvent consumption and extraction time, although concerns remain regarding energy consumption and the potential degradation of thermolabile compounds [93].

PLE is a method in which pressure is applied during the extraction process, enabling the use of temperatures higher than the boiling point of solvents. The application of high temperatures enhances mass transfer and accelerates the extraction process, making PLE typically more efficient than the conventional methods, with shorter extraction times and limited organic solvent consumption [90]. PLE is a technique that was effectively utilized for extracting glycoalkaloids from PP; in fact, Hossain et al. [57] found that a higher yield was obtained from PP using PLE (1.92 mg g^−1^ DW) compared to conventional SLE, which yielded 0.981 mg g^−1^ DW. In contrast, PLE is not widely used for extracting phenolic compounds due to the high temperatures, which can cause degradation of these heat-sensitive compounds. For instance, Wijngaard et al. [90] showed that TPC in ethanolic SLE extracts was higher (3.94 mg_GAE_ g^−1^ DW) than PLE (3.68 mg_GAE_ g^−1^ DW), highlighting that a careful optimization of temperature and extraction time in PLE is required to prevent the loss of bioactivity compounds during the process [96]. While PLE reduces solvent usage and accelerates extraction, the requirement for high-pressure systems increases operational costs and limits scalability [97].

SFE operates with supercritical fluids that surpass critical pressure and temperature, combining liquid and gas properties, which enhances efficiency and solvent penetration into food matrices. Carbon dioxide is commonly used due to its safety, low toxicity, and suitable critical point (31.6 °C; 7.386 MPa), preserving thermosensitive bioactive compounds [98,99,100]. SFE typically requires dried sample because the high-water content in vegetable matrices can compete with the solute and interact with the solvent, reducing the yields [83]. Few studies have applied SFE to PP. Lima et al. [44] recovered phenolic compounds from PP using SFE-CO_2_ at 80 °C, 350 bar, 20% (*v*/*v*) methanol, and a flow rate of 18.0 g min^−1^, achieving a 37% total recovery, with caffeic acid reaching 82%. They confirmed the valuable exploitation of water–organic solvent mixtures to enhance the recovery of phenolic acids like caffeic and chlorogenic acids. Although SFE is efficient, it generally produces lower phenolic extraction yields from vegetable matrices compared to UAE and MAE. These methods are more effective in breaking down cell structures, facilitating compound release, whereas SFE’s supercritical solvent offers good solvation and diffusion, but with less disruption of the cell matrix [44]. Despite being environmentally friendly and solvent-free, the high pressure and temperature requirements make SFE an expensive technique, primarily suitable for high-value bioactive compounds [101,102]. However, the efficiency of the solvent and the near-complete recovery of CO_2_ can help reduce operating costs in the long term.

In Table 6, the main findings regarding the application of non-conventional extraction methods for bioactive compound recovery from PP are reported.

**Table 6 foods-14-01333-t006:** Total phenolic content (TPC) and total glycoalkaloid content (TGA) obtained using non-conventional extraction methods (UAE, MAE, PLE, and SFE).

**Phenolic Compounds**				
**Extraction Method**	**Double Extraction**	**Operating Conditions**	**TPC (mg** **_GAE_/g db)**	**Reference**
UAE	-	80% methanol; ratio 1:10 (*w*/*v*); t: 30–900 min; T: 30–45 °C; f: 42 kHz	7.67 ± 0.79 (var. *Lady Rosetta*) 3.80 ± 0.09 (var. *Lady Claire*)	[48]
UAE	-	80% methanol (*v*/*v*); ratio 1:10 (*w*/*v*); t: 30–900 min; T: 30–45 °C; f: 33 kHz	4.24 ± 0.09 (var. *Lady Claire*)	[48]
UAE	-	50% methanol (*v*/*v*); ratio 1:20 (*w*/*v*); t: 30 min; T:25 °C; f: 40 Hz	9.09 *	[86]
UAE	SLE: methanol; t: 16 h; T: 22 °C ± 2; ratio 1:10 (*w*/*v*)	t: 20 min	3.26 ± 0.00	[103]
UAE	SLE: 80% (*v*/*v*) methanol; t: 16 h; T: 22 °C ± 2; ratio 1:10 (*w*/*v*)	t: 20 min	3.78 ± 0.00	[103]
UAE	SLE: ethanol; t: 16 h; T: 22 °C ± 2; ratio 1:10 (*w*/*v*)	t: 20 min	2.61 ± 0.00	[103]
UAE	SLE: 80% ethanol (*v*/*v*); t: 16 h; T: 22 °C ± 2; ratio 1:10 (*w*/*v*)	t: 20 min	3.26 ± 0.00	[99]
UAE	SLE: water; t: 16 h; T: 22 °C ± 2; ratio 1:10 (*w*/*v*)	t: 20 min	1.91 ± 0.00	[103]
UAE	SLE: methanol/water/acetic acid (70:25:5%, *v*/*v*); t: 60 min; L/S: 0.025	t: 20 min	n.d. (var. *Russet*)	[69]
UAE	SLE: ethanol/water/acetic acid (67:24:9%, *v*/*v*); t: 60 min; L/S: 0.025	t: 20 min	n.d. (var. *Russet*)	[69]
UAE	SLE: ethanol/water/acetic acid (51:46:3%, *v*/*v*); t: 60 min; L/S: 0.025	t: 20 min	n.d. (var. *Russet*)	[69]
MAE	-	67.33% methanol (*v*/*v*); t: 15 min; T: 25 °C; 1:20 S-L; power: 14.67%	3.94 ± 0.21 (var. *Russett Burbank*)	[94]
MAE	-	Methanol/water (30:70%, *v*/*v*) (100:0%); t: 5–15 min; L/S: 0.02; power: 63–229 W	3.92 *	[18]
MAE	-	Ethanol/water (20:80%, *v*/*v*) (80:20%,*v*/*v*); t: 2–8 min; T: 50–80 °C; L/S: 0.04–0.055; power: 300 W	9.8 *	[18]
PLE	-	70% ethanol (*v*/*v*); T: 125 °C	3.68 ± 0.0 (var. *Lady Claire*)	[90]
SFE	-	CO_2_; T: 80 °C; pressure: 350 bar; MeOH 20% (*v*/*v*); flow rate 18 g/min	2.24 mg TPC/g dry peels (var. *Maris Piper*)	[44]
**Glycoalkaloids**				
**Extraction Method**	**Double Extraction**	**Operating Conditions**	**TGA (mg/kg db)**	**Reference**
UAE	-	Solvent: water/acetic acid/sodium metabisulfite (95:4.5:0.5%, *v*/*v*); t: 5–25 min; T:25–85 °C; L/S: 0.025; amplitude: 50–100%	40.9–438	[91]
UAE	-	Methanol; t: 17 min; L/S: 0.01; amplitude: 61 μm	1102 *	[104]
UAE	-	Methanol; t: 2.5 min; ratio 1:10 *w*/*v*	1600 *	[79]
UAE	SLE: methanol/water/acetic acid (70:25:5%, *w*/*w*); t: 60 min; L/S: 0.025	t: 20 min	n.d. (var. *Russet*)	[69]
UAE	SLE: ethanol/water/acetic acid (67:24:9%, *w*/*w*); t: 60 min; L/S: 0.025	t: 20 min	n.d. (var. *Russet*)	[69]
UAE	SLE: ethanol/water/acetic acid (51:46:3%); t: 60 min; L/S: 0.025	t: 20 min	n.d. (var. *Russet*)	[69]
MAE	UAE (acetonitrile/formic acid = 99:1, *v*/*v*)/ water = 5:1, *v*/*v*; t: 10.70 min; power: 505 W; ratio 1:18 g mL^−1^	t: 6.10 min; power: 900 W	292.91 ± 3.90	[95]
PLE	-	Methanol/water (89:11%, *v*/*v*); t: 5 min preheating + 5 min static period; T: 80 °C; pressure: 1500 psi	1.92 * mg g^−1^	[57]

*: Standard deviation not reported; (-): not applicable. L/S: liquid/solid ratio in L/g; UAE: ultrasound-assisted extraction; MAE: microwave-assisted extraction; PLE: pressurized liquid extraction; SFE: supercritical fluid extraction.

## 4. Biological Activities of PP

Several studies have investigated the biological activity of PP, including its antioxidant, anti-inflammatory, antimicrobial, and antidiabetic potential, which is largely related to its phytochemical composition [22,23,78,83]. The biological effects of PP have been explored in various in vitro and in vivo models, demonstrating their potential application in the prevention and management of chronic diseases [23,105]. Additionally, recent research focused on their role in modulating cellular pathways related to oxidative stress, inflammation, and microbial growth, further reinforcing their functional value in health-related applications [73,106].

### 4.1. Antioxidant Activity

The potential of PPs as antioxidants in biological systems has attracted considerable scientific interest. As shown in Table 7, numerous studies focused on their remarkable antioxidant properties. In particular, PP extracts often exhibited significantly higher radical scavenging activity compared to flesh extracts [75,107]. Antioxidant capacity was assessed using various in vitro assays, including the scavenger activity in organic environment against 2,2-diphenyl-1-picrylhydrazyl (DPPH) radicals [75,78,103,108,109,110], as well as ferric reducing antioxidant power (FRAP) [78,105], and oxygen radical absorbance capacity (ORAC) [75]. Interestingly, some organic samples exhibited superior antioxidant performances compared to the non-organic counterparts. However, these differences were not always statistically significant [78]. Additionally, among different PP varieties, those with purple pigmentation demonstrated relevant antioxidant activity [75]. For instance, Sampaio et al. [111] evaluated the antioxidant activity of ten colored PP extracts and observed that the *Rosemary* variety showed the highest activity. Specifically, in the thiobarbituric acid-reactive substances (TBARS) assay, the *Rosemary* extract required the lowest concentration (IC_50_ = 26 μg mL^−1^) to reduce lipid peroxidation by 50%, outperforming the Trolox control [111]. Similarly, a study by Ansari et al. [110] investigated the antioxidant potential of the *Lady Rosetta* (LR) variety, characterized by red skin, and the *FT-1533* variety, with brown skin. Their findings highlighted that LR exhibited higher antioxidant activity, attributed to its greater phenolic compound content. These results align with previous studies such as that of Franková et al. [62], where the red-skinned *Cecil* variety displayed among the highest phenolic contents, along with the purple-skinned *Violet Queen* variety.

PPs have also been shown to exert protective effects in biological systems. Studies demonstrated their ability to shield red blood cells from oxidative stress induced by FeSO_4_ and ascorbate [112], and to mitigate cholesterol oxidation product (COP) toxicity in rats [113]. In the latter case, PP supplementation significantly increased antioxidant enzyme levels and hepatic glutathione, suggesting a role in preserving oxidative balance [113]. These findings collectively underscore the relevance of PPs as promising natural antioxidants, with potential applications in functional foods and nutraceuticals.

### 4.2. Anti-Inflammatory Activity

The bioactive compounds in PP were also proven to possess notable anti-inflammatory properties. Verma et al. [114] assessed the anti-inflammatory potential of ethanol PP extracts in a carrageenan-induced rat paw edema model, observing a dose-dependent reduction in swelling, with 200 mg kg^−1^ being the most effective. Expanding on these findings, Wahyudi et al. [115] tested hydroethanolic PP extracts both in vitro and in vivo. It has been found that in vitro, the extract inhibited the nitric oxide production in RAW 264.7 macrophages (IC_50_ = 141 μg mL^−1^) while in vivo, it reduced paw swelling at doses of 100, 200, and 400 mg kg^−1^, with diclofenac as the positive control. The extract also alleviated pain in male Wistar rats, with effects comparable to paracetamol at 100 and 200 mg kg^−1^. Sampaio et al. [111] further investigated the anti-inflammatory activity of PP, focusing on phenolic compounds from the *Rosemary* variety. Their extract significantly inhibited RAW 264.7 macrophage growth (IC_50_ = 141 μg mL^−1^). While most studies focused on phenolics, Kenny et al. [116] investigated the anti-inflammatory effects of glycoalkaloids in PP, emphasizing the critical role of their aglycone unit. As nitrogenous analogs of steroidal saponins like diosgenin, glycoalkaloids demonstrated significant efficacy in reducing inflammatory responses. Notably, they suppressed the production of pro-inflammatory cytokines in stimulated leukemia T lymphoblasts and attenuated nitric oxide generation in a RAW 264.7 macrophage inflammation model [116].

### 4.3. Antibacterial and Antiviral Activities

PPs have also demonstrated significant microbiological potential. A study conducted by Amanpour et al. [117] tested the antibacterial activity of PP extracts against *Streptococcus pyogenes* (*S. pyogenes*) PTCC 1447, *Staphylococcus aureus* (*S. aureus*) PTCC 1113, *Pseudomonas aeruginosa* (*P. aeruginosa*) PTCC 1430, and *Klebsiella pneumoniae* (*K. pneumoniae*) PTCC 1053, observing a stronger effect on Gram-positive bacteria compared to Gram-negative ones. Similarly, Gebrechristos et al. [118] used microdilution and diffusion tests to assess antimicrobial properties against *Escherichia coli* (*E. coli*), *Salmonella enterica* (*S. enterica*), and *S. aureus* (Minimum Inhibitory Concentrations (MICs): 7.5 ± 2, 5.8 ± 2, and 4.7 ± 1 mg mL^−1^, respectively), while no activity was detected against *K. pneumoniae* and *Lysteria monocytogenes* (*L. monocytogenes*). HPLC analysis identified caffeic, chlorogenic, and neochlorogenic acids as the key antimicrobial compounds. Regarding antiviral properties, Silva-Beltrán et al. [72] found that acidified ethanolic PP extracts of *Fianna* variety reduced human enteric viral surrogates, specifically Avian virus 05 (Av-05) and MS2 bacteriophages. After a 3 h incubation with 5 mg mL^−1^ extracts, Plaque-Forming Units (PFU/mL) decreased by 2.8 and 3.9 log_10_, respectively, in a dose-dependent manner.

PP extracts also showed antiprotozoal features. Friedman et al. [92] tested the extracts from Russet, red, purple, and fingerling PPs against *Trichomonas vaginalis* and Trichomonas foetus, finding that the most effective were active at 10% (*w*/*v*), with *Russet* extracts being the most active. Among glycoalkaloids, *α*-chaconine and *α*-solanine exhibited significant antiparasitic properties, whereas their shared aglycone, solanidine, showed only a mild effect. Notably, *α*-solanine was considerably more potent than *α*-chaconine. In contrast, phenolic compounds, such as caffeic and chlorogenic acids, as well as quercetin displayed only modest responses [92].

### 4.4. Other Activities

PPs have also been investigated for their anti-obesity and antidiabetic potential, to offer possible alternative treatments of type 2 diabetes [22]. Bioactive compounds in PP, including glycoalkaloids and flavonoids, were proven to help glucose level regulation by preserving pancreatic β-cell function and inhibiting α-glucosidase, thereby reducing the glucose absorption via GLUT2 [106]. Previously, Arun et al. [61] explored in vitro the antidiabetic properties of PP extracts, by comparing the ability of different solvent fractions to inhibit α-glucosidase activity. The methanolic extract showed the strongest inhibition (IC_50_ = 184.36 μg mL^−1^) while the ethyl acetate extract, although slightly less effective (IC_50_ = 197.13 μg mL^−1^), enhanced glucose uptake in muscle and adipose cells via GLUT4 and reduced intracellular reactive oxygen species (ROS), suggesting broader metabolic benefits. PP also demonstrated promising potential in weight management. A study by Elkahoui et al. [47] showed that supplementing a high-fat diet with PP powder reduced weight gain in mice by up to 73%; the effect mostly related to *α*-solanine and *α*-chaconine content. Additionally, previous research found that these compounds, along with phenolics, contribute to weight reduction by affecting gut microbiota and metabolism [119,120,121].

Another notable benefit of PP is linked to anti-aging effects. The PP extract stimulates type I collagen synthesis in fibroblasts by activating the PI3K (Phosphoinositide 3-kinase)/Akt (Protein Kinase B) and MAPK (Mitogen-Activated Protein Kinase)/ERK (Extracellular signal-Regulated Kinase) signaling pathways through TGF-β (Transforming Growth Factor-beta) receptors [106]. Previous studies also demonstrated that compounds such as ascorbic acid, phenolic compounds, and flavonoids can promote the synthesis of type I collagen in the dermis [122].

PPs have also shown benefits against atopic dermatitis in animal models. In a study on *Dermatophagoides farinae*-induced atopic dermatitis, the ethanol extract of PPs (*Jayoung* variety) reduced disease severity, lowered serum IgE (Immunoglobulin E) levels, and decreased mast cell infiltration while modulating Th2 (T helper 2) cytokine expression and restoring filaggrin levels in skin lesions [123].

Finally, PP extracts exhibited anticancer properties. Sampaio et al. [111] evaluated the antiproliferative and cytotoxic properties of hydroethanolic extracts from colored PP. The extracts showed significant activity against four human tumor cell lines (NCI-H460, HepG2, MCF-7, and HeLa), with GI_50_ (Growth Inhibition 50) values ranging from 49 to 365 μg mL^−1^. The *Rosemary* variety recorded the strongest effect, requiring the lowest concentrations for tumor inhibition. Cytotoxicity assays on non-tumor porcine liver cells (PLP2) confirmed the extracts’ safety (GI_50_ > 400 μg mL^−1^), except for the *Rosemary* extract, which displayed mild cytotoxicity at higher doses (GI_50_ = 304 μg mL^−1^).

**Table 7 foods-14-01333-t007:** The biological activities of PP extracts, including the type of extract, bioactive compounds identified, and the main results from the studies reviewed.

Activity	Extract Type	Bioactive Compounds	Main Results	Reference
Antioxidant activity in cell-based in vitro assays, anti-inflammatory effects in RAW 264.7 mouse macrophages, and antiproliferative activity against cancer cell lines	PP ethanol extract 80% (*v*/*v*)	Anthocyanins, (non-anthocyanins) phenolic compounds, flavonoids, caffeic, caffeoylquinic acids, kaempferol	*Rosemary* extract showed strong antioxidant (IC_50_ = 26 µg mL^−1^) and anti-inflammatory effects (IC_50_ = 141 µg mL^−1^). All extracts showed antiproliferative activity against NCI-H460, HepG2, MCF-7, and HeLa cell lines (GI_50_ 49–365 μg mL^−1^). The *Rosemary* variety exhibited the highest activity.	[111]
Antioxidant activity by DPPH (40 µL) and FRAP (0.1 µL) methods	PP ethanol extract	Phenolic and flavonoid compounds	*Lady Rosetta* exhibited superior antioxidant activity (RSA = 63.32) and nutritional content compared to FT1533, with doses of 40 µL for RSA and 0.1 µL for FRAP.	[110]
Antioxidant activity by DPPH method and anti-microbial against *Bacillus subtilis* ATCC 6633, *Staphylococcus aureus* ATCC 29213, *E. coli* ATCC 25922, *Salmonella t.* ATCC14028, and *C. albicans* ATCC 10231	Water, methanol, and ethanol (80 and 100%) extracts	Phenolic and flavonoid compounds (vanillic and hesperidin)	At a dose of 70 µL (200 and 400 ppm), PP methanol extract at 400 ppm was more effective in inhibiting Gram-positive and Gram-negative bacteria, as well as *Candida albicans*, compared to ampicillin (control).	[103]
Antioxidant and antiviral activities (bacteriophages Av-05 and MS2 and host *E. coli* ATCC 4076)	PP ethanol extract and water extract	Phenolic and flavonoid compounds (quercetin and gallic acid)	At concentrations of 1, 3, and 5 mg mL^−1^, phenol and flavonoid compounds from ethanol extract and their derivatives inhibited the replication of MS2 and Av-05 phage viruses (foodborne viruses) and exhibited antioxidant activity.	[72]
Antioxidant activity by DPPH method and nitric oxide activity	Ethyl acetate and methanol extract	Phenolic compounds (gallic acid)	The ethyl acetate extract had a higher phenolic content and better antioxidant capacity than the methanol extract, showing stronger DPPH and nitric oxide activity at doses of 0.2, 0.4, 0.6, 0.8, and 1 mg mL^−1^.	[108]
Antioxidant activity by DPPH radical, FRAP, and ABTS methods	PP extract with acetate acid 5%	Phenolic compounds, flavonoids, and glycoalkaloids	Doses of 0.8 µL, 10 µL, and 60 µL were used for each method. The antioxidant activity of organic PP compounds was higher than that of non-organic or conventional ones, though not significantly.	[78]
Antioxidant activity (DPPH, nitric oxide) and antidiabetic activity (L6 rat skeletal myoblast cell line)	Hexane, ethyl acetate, methanol extracts	Gallic, caffeic, ferulic, chlorogenic acids	Ethyl acetate extracts had the highest phenolic content and antioxidant activity. EYP showed the best α-glucosidase inhibition (IC_50_ = 197.13 µg mL^−1^), ROS scavenging, and glucose uptake in L6 cells. Young PP exhibited superior antioxidant and antidiabetic activity compared to mature PP, with effects observed at concentrations of 1, 10, and 100 µL mL^−1^.	[61]
Antioxidant activity by Trolox (TEAC), DPPH, and oxygen radical absorbance (ORAC) methods	Phenolic fractions	Polyphenolic compounds, anthocyanins	Purple PP exhibited the highest total phenolic content (7.2 mg g^−1^ peel) and antioxidant activity compared to other varieties.	[75]
Antioxidant activity in vitro with erythrocytes; protection against FeSO_4_ and ascorbic acid-induced oxidative damage; inhibition of H_2_O_2_-induced morphological changes in rat erythrocytes; protection of human erythrocyte membrane proteins from ferrous ascorbate-induced damage	Aqueous PP extract	Phenolic compounds	PP extract (2.5 mg mL^−1^) exhibited antioxidant activity by inhibiting lipid peroxidation (80–85% inhibition in both rat and human RBC systems), protected erythrocytes from oxidative damage, prevented H_2_O_2_-induced morphological changes in rat RBCs, and safeguarded human erythrocyte membrane proteins from oxidative damage induced by ferrous ascorbate.	[112]
Antioxidant activity, reduction in toxicity of COPs in vivo	Aqueous PP extract	Phenolic compounds	There was a significant increase in antioxidant enzyme activities and liver glutathione, with a decrease in liver enzymes, renal markers, and COP levels. A PPE diet at 2% and 3% doses effectively reduced COP toxicity in rats.	[113]
Analgesic activity with Wistar rats by placing at hotplate 60 °C, anti-inflammatory activity with Wistar rats induced by 1% carrageenan, anti-biofilm activity with streptococcus ATCC 25175	PP ethanol extract	Phenolic and flavonoid compounds	There were significant anti-inflammatory, anti-biofilm, and analgesic effects. The analgesic was tested at 50, 100, and 200 mg kg^−1^, and anti-inflammatory at 100, 200, and 400 mg kg^−1^. Anti-biofilm effects were assessed using 5%, 10%, and 20% extract (*w*/*w*).	[115]
Anti- inflammatory activity with Wistar rats induced by 0.1% carrageenan	PP 70% ethanol extract	Flavonoid and phenolic compounds	At doses of 100 and 200 mg kg^−1^, PP ethanol extract had an anti-inflammatory effect in rats with induced edema.	[114]
Anti-inflammatory activity with human Jurkat T cell and RAW 246.7 mouse macrophages by biomarkers IL2 and IL8	Ethyl acetate fraction and methanol fraction	Glycoalkaloid compounds	Glycoalkaloid compounds extracted from the PP fraction exhibited anti-inflammatory effects at doses of 5, 10, and 25 μg mL^−1^.	[116]
Antibacterial activity *with E. coli* ATCC 25922, *S. enterica* ATCC 1311, *K. pneumonia* ATCC 2473, *S. aureus* ATCC 6538, *L. monocytogenes* ATCC 19115, and antioxidant activity by DPPH method	PP ethanol extract	Phenol and flavonoid compounds	PP extract (10 mg mL^−1^) exhibited antibacterial activity (MIC of 7.5, 5.8, and 4.7 mg mL^−1^) and antioxidant properties, with its phenolic compounds inhibiting lipid oxidation and bacterial growth.	[118]
Anti-aging activity with NHDF cell	PP ethanol extract 50% (*v*/*v*)	Phenolic and flavonoid compounds	The mRNA expression of COL 1A1 and COL 1A2 increased, promoting collagen synthesis via the Akt/PI3K and MAPK/ERK signaling pathways, with doses of 5, 10, and 15 µg mL^−1^ of ethanol extract.	[122]
Antibacterial activity with (*S. pyogenes* PTCC 1447, *S. aureus* PTCC 1113, *P. aeruginosa* PTCC 1430, *K. pneumonia* PTCC 1053)	PP ethanol extract 80% (*v*/*v*)	Flavonoid compounds	A dose of 30 μL of PP extract, containing phenolic compounds, anthocyanins, and flavonoids, exhibited antibacterial effects, particularly against Gram-positive bacteria.	[117]
Antitrichomonad activity against pathogenic strains (*Trichomonas vaginalis* and *Tritrichomonas foetus*)	PP extract (ethanol 50%/acetic acid 0.5%)	Glycoalkaloids, phenolic compounds	PP extracts inhibited the growth of three trichomonad species at a 10% *w*/*v* concentration, with Russet samples being the most effective. No inhibitory effect on normal native vaginal bacteria species was observed.	[92]
Anti-obesity and antidiabetic activity	PP powder	Glycoalkaloids (α-chaconine, α-solanine), phenolic compounds	PP powder supplementation (10–20%) in high-fat diets reduced weight gain in mice by 17–73%, with reductions in epididymal white adipose tissue (22–80%). Weight gain was negatively correlated with glycoalkaloid content. Changes in microbiota and obesity-associated genetic biomarkers were observed.	[47]
Anti-atopic dermatitis	PP ethanol extract	Phenolic compounds	PP reduced dermatitis severity, lowered IgE levels, decreased mast cell infiltration, modulated Th2 cytokines, restored filaggrin levels.	[123]

PPs: Potato peels; PPE: potato peel extract; DPPH: 2,2-diphenyl-1-picrylhydrazyl; FRAP: ferric reducing antioxidant power; ABTS: 2,2′-Azino-bis (3-ethylbenzothiazoline-6-sulfonic acid) diammonium salt; ORAC: oxygen radical absorbance capacity; RSA: radical scavenging assay; COP: cholesterol oxidation products; NHDF: normal human dermal fibroblasts.

## 5. Application of PP in Food Industry

The PP’s high nutritional value suggested its exploitation as a functional ingredient in food formulations [42]. The incorporation of PP into food products can be achieved in different forms, including powder, extract, or isolated fiber. With consumers’ rising awareness about functional and health-promoting foods, there is a growing interest among both researchers and food industries to improve food formulations with natural bioactive compounds from plant sources [82]. In this context, the research on PP-enriched food products and their main findings are collected in Table 8.

### 5.1. Cereal-Based Products

The incorporation of PP in cereal-based products has been investigated for its effects on dough properties and final product characteristics, including functional, technological, and sensory attributes. Curti et al. [124] analyzed the addition of PP fiber (0.4 g fiber/100 g flour) in bread formulations to assess its impact on both texture and shelf-life. The addition of PP fiber increased the water-retention capacity, resulting in a softer crumb over seven days of storage. At the end of the storage period, the crumb hardness of PP-fortified bread was lower compared to the control formulation (3.7 ± 0.6 N vs. 4.5 ± 0.7 N), indicating reduced starch retrogradation and improved textural stability. Crawford et al. [125] investigated the inclusion of PP powder (5% *w*/*w*) from the *Russet* variety in quinoa flatbreads: the PP-enriched formulation exhibited modified color attributes due to the natural pigments present in the peel, leading to darker tones with increased a* values (red hue) and lower L* values (lightness); additionally, the addition of PP reduced acrylamide content in the final product. Jacinto et al. [126] studied the enrichment of PP flour (PPF) into gluten-free bread, demonstrating that a formulation containing 5% (*w*/*w*) PPF enhances both sensory acceptability and nutritional profile, increasing mineral, lipid, and protein content.

Ben Jeddou et al. [127] incorporated PPF (2–10% *w*/*w*) of the *Spunta* variety in cakes and dough, observing significant color variations with decreased yellowness and lightness. The addition of PP improved dough strength and reduced cake hardness, with the 5% (*w*/*w*) PP formulation lowering hardness by 30.2% compared to the control. PP’s water-binding and fat absorption capacities contributed to improve texture, while sensory analysis showed no significant impact on consumer acceptance despite color changes. Similarly, Hallabo et al. [130] partially replaced wheat flour with PP powder in biscuit formulations, resulting in higher water absorption, prolonged dough development time, and enhanced dough stability. This incorporation also led to higher fiber content and improved antioxidant activity, further demonstrating PP’s functional potential in baked goods.

Finally, Fradinho et al. [131] explored the incorporation of PP extract, obtained through subcritical water extraction, into gluten-free pasta. Their findings highlighted improvements in antioxidant activity and phenolic content, as well as positive effects on the pasta’s nutritional composition and cooking quality.

### 5.2. Vegetable Oils

Lipid oxidation compromises the stability of vegetable oils and processed foods, leading to rancidity, reduced shelf-life, and the formation of harmful oxidation products [22]. To mitigate these effects, synthetic antioxidants such as butylated hydroxytoluene (BHT) and butylated hydroxyanisole (BHA) are commonly exploited [22]. However, concerns over their potential health risks, including carcinogenicity, have driven growing interest in natural alternatives [43], which are preferred for their health benefits and lack of adverse effects [21]. In this context, PP emerged as a sustainable antioxidant source for the food industry [145] with extracts exhibiting properties comparable, or even superior to, synthetic alternatives [138]. Their application in vegetable oils and processed foods not only enhances oxidative stability but also aligns with consumer demand for clean-label products and sustainable resource utilization [145]. Mohdaly et al. [136] compared the efficacy of PP and sugar beet pulp extracts in sunflower and soybean oils stored at 70 °C for 72 h. At 200 ppm, PP extracts demonstrated superior antioxidant activity compared to BHT and BHA, but were less effective than tert-butylhydroquinone (TBHQ). Amado et al. [50] optimized the extraction of the antioxidant compounds, particularly chlorogenic and ferulic acids, from PP to minimize oxidative indices in soybean oil under the same storage conditions. The PP extracts reduced peroxide value (PV) (22.8% inhibition), total oxidation value (22.4% inhibition), and p-anisidine value (*p*-AnV) (19.3% inhibition). Franco et al. [135] further demonstrated that ethanolic PP extracts significantly reduced lipid oxidation in soybean oil stored under accelerated conditions (60 °C for 15 days). The study assessed three extract concentrations (14.01, 20.37, and 31.94 ppm chlorogenic acid equivalent), with the highest concentration achieving a 52% inhibition of conjugated diene formation, surpassing the 41% inhibition observed with BHT. More recently, Saeed et al. [137] validated previous findings, showing that PP extract at 3200 ppm effectively inhibited lipid oxidation in sunflower oil over 60 days at 25 °C, exhibiting antioxidant activity comparable to BHA. The extract limited increases in oxidation indices, including PV, *p*-AnV, and iodine value (IV), while preventing fatty acid degradation and rancidity. After 60 days, oil containing PP extract exhibited lower free fatty acids (0.102%), PV (7.00 meq kg^−1^), and *p*-AnV (0.55%) compared to the control, while maintaining a higher IV (82 vs. 75 for BHA). These findings highlighted the potential of PP extract as a natural alternative for enhancing the oxidative stability of vegetable oils.

### 5.3. Meat and Fish Products

The incorporation of PP was also investigated in the meat and fish sector, focusing on its potential to enhance product quality, shelf-life, and nutritional value. Kanatt et al. [139] demonstrated that adding PP extract to irradiated lamb meat significantly reduced lipid peroxidation, as evidenced by lower thiobarbituric acid (TBA) values and carbonyl content. PP extract (0.04% *w*/*w*) exhibited antioxidant activity comparable to BHT, inhibiting oxidative rancidity without affecting the meat’s flavor or aroma. These findings suggested that PP extract can significantly enhance the storage quality of radiation-processed meat, offering a natural alternative to synthetic antioxidants and highlighting its potential as a functional ingredient in meat preservation. Saed and El-Waseif [140] found that PP powder acted as a functional fat replacer in low-fat beef meatballs, improving nutritional composition and texture. The inclusion of PP powder increased crude dietary fiber, protein, ash content, water-holding capacity, and cooking yield, while reducing cooking loss and energy intake by 24%. The increased water-holding capacity contributed to a juicier texture, while maintaining sensory acceptance even at 50% fat replacement. Additionally, Espinoza-García et al. [142] investigated the integration of PP powder (2–10% *w*/*w*) in pork patties, highlighting improvements in oxidative stability and physicochemical properties during refrigerated storage. The addition of PP reduced pH fluctuations, lipid oxidation, and cooking loss while enhancing water-holding capacity. Although some textural modifications were observed, sensory evaluations of cooked patties showed no significant differences in juiciness, texture, or overall acceptability compared to control samples.

Furthermore, the application of PP in seafood products was also explored. Sabeena Farvin et al. [87] investigated the antioxidant potential of PP extracts in minced horse mackerel, using water and ethanol extracts from the *Sava* variety at 2.4 and 4.8 g kg^−1^. Samples were stored at 5 °C for 96 h, and ethanolic extracts proved the most effective in delaying lipid and protein oxidation, reducing peroxide values, volatile compounds, and carbonyl formation. They also helped preserve α-tocopherol and essential amino acids. In contrast, aqueous extracts showed lower efficacy, possibly due to reduced phenolic content or the pro-oxidative effects of certain co-extracted compounds. Similarly, Albishi et al. [75] evaluated the antioxidant activity of PP extracts from four potato varieties (*Russet*, *Innovator*, *Purple*, and *Yellow*) in cooked salmon stored for seven days. The extracts inhibited lipid oxidation, as measured by TBARS levels, with *Russet* PP showing the highest effectiveness (83.4% inhibition), followed by *Purple* (39.7%), *Innovator* (31.4%), and *Yellow* (9.48%). The results highlighted the superior antioxidant capacity of *Russet* PP, even outperforming BHA, reinforcing the potential of PP extracts as natural preservatives in fish-based products.

### 5.4. Dairy Products

Currently, limited studies examine PP incorporation into dairy products, highlighting a research gap in this field. Brahmi et al. [82] reported that adding PP extract to yogurt enhances its antioxidant capacity and sensory properties. The PP extract increased the total phenolic and flavonoid content, boosting antioxidant capacity. The fortification also improved physicochemical attributes, such as total solids and acidity, while sensory analysis revealed a positive impact on yogurt consistency and overall acceptability. Salama et al. [144] evaluated the potential of PP powder as a functional ingredient in processed cheese formulations. While its incorporation improved protein content, mineral composition, and antioxidant activity, it also modified key textural and melting properties. Low concentrations (1–2% *w*/*w*) improved the nutritional profile without significantly compromising sensory acceptability, whereas higher levels (4% *w*/*w*) negatively affected texture and consumer preference. Teklehaymanot et al. [143] investigated the antioxidant capacity of phenolic extracts from PP and their efficacy in preserving cow butter compared to synthetic antioxidants. Butter treated with 0.3% (*w*/*w*) PP extract at 20 °C showed lower peroxide and free fatty acid levels, reduced microbial growth, and improved oxidative stability. The Rancimat test confirmed a shelf-life extension to 165 days, compared to only 5 days for untreated butter. These findings demonstrated the viability of PP extracts as a cost-effective, health-promoting addition to dairy products and a sustainable, effective, natural alternative to synthetic preservatives in lipid-based foods.

## 6. Conclusions

The potato processing industry generates substantial waste, with PP being a major by-product. It is anticipated that there will be a substantial increase in its production in the forthcoming years, which will result in a concomitant rise in greenhouse gas emissions. Its valorization has gained attention as a sustainable strategy to reduce waste and improve resource efficiency. PP is a rich source of bioactive compounds, including polyphenols, dietary fiber, and minerals, with promising applications in the food and nutraceutical industries.

Among extraction techniques, UAE has proven particularly efficient, but also, other emerging technologies such as MAE and PLE show promise in enhancing yields and reducing solvent consumption. Future research should focus on optimizing eco-friendly extraction processes to improve efficiency, scalability, and sustainability. Additionally, PP bioactive compounds exhibit antioxidant, anti-inflammatory, and antimicrobial properties, though their bioavailability and bioaccessibility remain key challenges requiring further investigation through preclinical and clinical studies.

PP incorporation in food products offers advantages such as improved nutritional quality, oxidative stability, and sensory attributes, making it a potential alternative to synthetic additives. Its application has been explored in bakery products, vegetable oils, meat, fish, and dairy, with positive impacts on texture, color, and overall nutritional value. However, variability in PP composition due to potato variety, cultivation, and processing conditions poses challenges for standardization and industrial implementation.

Despite its potential, concerns remain regarding glycoalkaloids, which require effective detoxification strategies due to toxicity risks. However, recent evidence suggests that, at controlled doses, they may offer health benefits. Future research should refine extraction methods, assess compound stability, and evaluate glycoalkaloid dose–response relationships to ensure their safe application. By addressing these challenges, the valorization of PP can contribute to a more sustainable and circular food industry while promoting innovative functional ingredients with potential health benefits.

## Figures and Tables

**Figure 1 foods-14-01333-f001:**
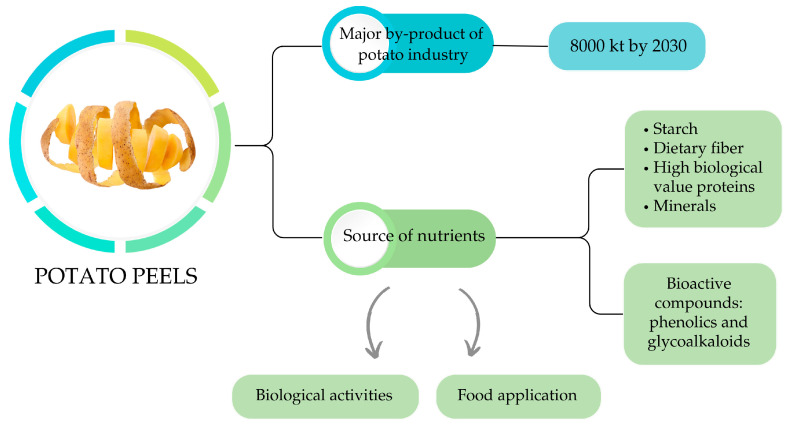
Potato peel production and characteristics. Source: authors’ elaboration.

**Figure 2 foods-14-01333-f002:**
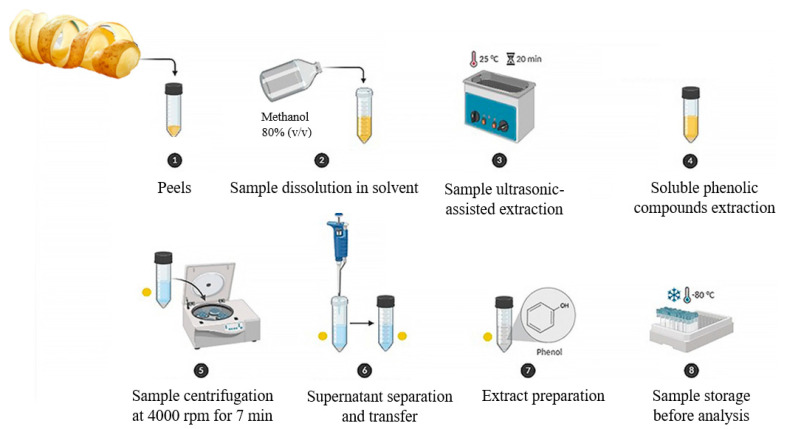
A schematic representation of the extraction process for phenolic compounds. Source: authors’ elaboration, created with BioRender.

**Table 3 foods-14-01333-t003:** The contents of α-chaconine, α-solanine, and total glycoalkaloids (TGAs) in PP as reported in the literature.

Potato Peel Variety	*α*-Chaconine	*α*-Solanine	TGA	Reference
*Bintjie* (µg/g DW)	1047 ± 188	551 ± 139	1597	[74]
*Kennebec* (µg/g DW)	1833 ± 139	1750 ± 272	3583	[74]
*Mona Lisa* (µg/g DW)	1051 ± 223	535 ± 172	1586	[74]
*Spunta* (µg/g DW)	1021 ± 46.4	446 ± 22.2	1467	[74]
*Vitelotte* (µg/g DW)	1762 ± 228	980 ± 156	2742	[74]
*Lady Rosetta* (µg/g DW)	1358 ± 63	1012 ± 109	2370	[74]
*Conventional gold* (μg/g of powder weight)	670 ± 130	253 ± 44	920 ± 140	[78]
*Conventional red* (μg/g of powder weight)	1297 ± 56	412 ± 24	1709 ± 61	[78]
*Conventional Russet* (μg/g of powder weight)	424 ± 30	215 ± 43	639 ± 52	[78]
*Organic gold* (μg/g of powder weight)	2830 ± 370	750 ± 120	3580 ± 390	[78]
*Organic red* (μg/g of powder weight)	610 ± 110	239 ± 34	850 ± 120	[78]
*Organic Russet* (μg/g of powder weight)	1180 ± 110	374 ± 54	1550 ± 120	[78]
*Red potato* (μg/g)	1604 *	572 *	2180 ± 170	[47]
*Gold potato* (μg/g)	1301 *	636 *	1940 ± 170	[47]
*Organic Russet* (μg/g)	593 *	268 *	861 ± 10	[47]
*Non-organic Russet* (μg/g)	781 *	347 *	1128 ± 1	[47]
*Desirèè* (mg/g FW)	118.81 ± 4.77	52.94 ± 4.98	171.75 ± 6.57	[80]
*Agria* (mg/kg DW)	251 ± 41	234 ± 32	485 ± 72	[79]
*Mona Lisa* (mg/kg DW)	892 ± 29	306 ± 19	1198 ± 40	[79]
*Vivaldi* (mg/kg DW)	117 ± 9	143 ± 9	260 ± 8	[79]
*Amandine* (mg/kg DW)	1742 ± 8	1081 ± 27	2823 ± 33	[79]
*Valfi* (mg/100 g DM)	58.2 ± 0.2	24.9 ± 0.2	83.1 ± 0.2	[81]
*Blaue Elise* (mg/100 g DM)	45.5 ± 0.3	21.2 ± 0.1	66.6 ± 0.2	[81]

*: Standard deviation not reported; DW: dry weight; FW: fresh weight; DM: dry matter.

**Table 4 foods-14-01333-t004:** Total phenolic content (TPC, mg GAE/g dry weight) obtained through SLE, as reported in the literature.

Solvent	Operating Conditions	TPC (mg_GAE_/g DW)	Reference
Methanol	t: overnight; T: ~25 °C; ratio 1:10 (*w*/*v*)	2.91 ± 0.02 (var. *Diamond*)	[84]
Ethanol	t: overnight; T: ~25 °C; ratio 1:10 (*w*/*v*)	2.74 ± 0.03 (var. *Diamond*)	[84]
Acetone	t: overnight; T: ~25 °C; ratio 1:10 (*w*/*v*)	2.39 ± 0.04 (var. *Diamond*)	[84]
Hexane	t: overnight; T: ~25 °C; ratio 1:10 (*w*/*v*)	1.12 ± 0.04 (var. *Diamond*)	[84]
Diethyl ether	t: overnight; T: ~25 °C; ratio 1:10 (*w*/*v*)	1.12 ± 0.03 (var. *Diamond*)	[84]
Petroleum ether	t: overnight; T: ~25 °C; ratio 1:10 (*w*/*v*)	1.08 ± 0.04 (var. *Diamond*)	[84]
Ethanol/acetic acid (95:5%, *v*/*v*)	t: 72 h; L/S: 0.01	14.031 ± 1.881 (var. *Fianna*)	[72]
Water	t: 72 h; L/S: 0.01	4.160 ± 0.974 (var. *Fianna*)	[72]
Methanol/water (50:50%, *v*/*v*)	t: 60 min; T: 25 °C; ratio 1:20 (*w*/*v*)	6.26 *	[86]
Ethanol/water (20:80%, *v*/*v*)-(90:10%)	t: 10–150 min; T: 23 °C; L/S: 0.01–0.09	9.68 a *	[18]
Methanol	t: 30 min; T: 75 °C; L/S: 0.05	3.33 a *	[18]
Ethanol	t: overnight; T: 5 °C; L/S: 0.01	2.5 a *	[18]
Methanol and 75% ethanol	t: 22 min; T: 80 °C	1.26–3.94	[83]
Methanol/water (80:20%, *v*/*v*)	t:15 h; T:23 °C	2.17 ± 0.02 (var. *Lady Claire*)	[48]
Methanol/water (80:20%, *v*/*v*)	t:15 h; T:23 °C	3.28 ± 0.07 (var. *Lady Rosetta*)	[48]
Ethanol/water (80:20%, *v*/*v*)	t: 150 min; ratio 1:30 (*w*/*v*)	204.41 ± 8.64 mg_GAE_/100 g DW	[82]
Ethanol (36.2–100%, *v*/*v*)	t: 5–150 min; T: 25–90 °C; ratio 1:20 (*w*/*v*)	3.2–10.3 * mg_GAE_/100 g DB (var. *Agria*)	[50]
Ethanol	t: overnight; T: ~25 °C; ratio 1:10 (*w*/*v*)	68.7 * mg_GAE_/100 g DB (var. *Sava*)	[87]
Water	t: overnight; T: ~25 °C; ratio 1:20 (*w*/*v*)	26.1 * mg_GAE_/100 g DB (var. *Sava*)	[87]
Water	t: 24 min; T: 25 °C; L/S: 0.02	2.51 *	[88]
Ethanol/water (10:90%)	t: 24 h; T: 27 °C; ratio 1:20 (*w*/*v*)	3.95 ± 0.02 (var. organic yellow potatoes “*Bologna*”)	[89]
Water	t: 24 h; T: 27 °C; ratio 1:20 (*w*/*v*)	2.92 ± 0.41 (var. organic yellow potatoes “*Bologna*”)	[89]
Ethanol/water (75:25%)	t: 22 min; T: 80 °C	3.94 ± 0.01 (var. *Lady Claire*)	[90]

*: Standard deviation not reported; L/S: liquid/solid ratio in L/g; a: mg chlorogenic acid equivalents/g; DW: dry weight; DB: dry basis; GAE: gallic acid equivalent.

**Table 5 foods-14-01333-t005:** Total glycoalkaloid content (mg kg^−1^ DB) obtained through SLE, as reported in the literature.

Solvent	Operating Conditions	TGA (mg/kg DB)	Reference
Water/acetic acid/sodium metabisulfite (95:4.5:0.5%, *v*/*v*)	t: 5.5–69.5 min; T: 25.5–89.5 °C; L/S: 0.025	42.1 ± 0.8–630.1 ± 70.2 (var. *Rooster*)	[91]
Ethanol/water/acetic acid (50:49.5:0.5%, *v*/*v*)	t: 60 min; L/S: 0.01	1200–5300 *	[92]
Methanol/water/acetic acid (80:19.5:0.5%, *v*/*v*)	t: 30 min; T: 4 °C; L/S: 0.01	1026–5342 *	[74]
Methanol	t: 17 min; T: 23 °C; L/S: 0.01	711 *	[18]
Methanol/water (90:10%, *v*/*v*)	t: 60 min; T: 23 °C; L/S: 0.01	0.981 *	[57]
0.02 M heptanesulfonic acid in 1% (*v*/*v*) aqueous acetic acid	t: -; T: 25 °C; ratio 1:10 (*w*/*v*)	171.75 ± 6.57 (mg/g FW–var. *Desirèè*)	[80]

* Standard deviation not reported; L/S: liquid/solid ratio in L/g; DB: dry basis.

**Table 8 foods-14-01333-t008:** A summary of food products incorporating PP and the observed effects on their properties.

Food Industry	Food Product	% PP	Effects of PP Incorporation	Reference
**Cereal-based products**	Bread (+PP fiber)	0.4% (*w*/*w*) PP fiber	Reduced staling by maintaining moisture, increasing frozen water content, and improving softness, even at low levels of incorporation	[124]
	Flatbread	5% (*w*/*w*) PP powder	Modified color attributes, resulting in darker tones with increased a* values and lower L* values, and reduced acrylamide content	[125]
	Gluten-free bread	2.5, 5.0, 7.5% (*w*/*w*) PPF	Improved nutritional profile by increasing mineral, lipid, and protein content, supporting a balanced diet for individuals with celiac disease	[126]
	Cake	2–10% (*w*/*w*) PPF	Improved dough texture, reduced cake hardness, increased dough strength and elasticity, and enhanced fiber and protein content	[127]
	Cake	4, 6, 8% (*w*/*w*) PPF	Increased nutrient retention, total phenolic and antioxidant content in cabinet-dried PP flour, enhanced cake quality and acceptability at 4% incorporation, and provided a sustainable alternative for cereal products	[128]
	Biscuits (+PP fiber)	5, 10, 15% (*w*/*w*) PP fiber	Improved nutritional profile by increasing dietary fiber, carbohydrates, ash, and fat content, while also influencing sensory properties such as color and overall acceptability	[129]
	Biscuits	6% (*w*/*w*) of flour weight	Increased water absorption, prolonged dough development time, enhanced dough stability, higher fiber content, and improved antioxidant activity	[130]
	Gluten-free pasta	4% (*w*/*w*) PP extract	Increased phenolic content and antioxidant activity while maintaining good texture and color	[131]
	Cracker	5% (*w*/*w*) PP powder	Increased dietary fiber, phenolic content, and antioxidant activity, enhancing the nutritional and functional properties of crackers	[132]
	Chips	2, 4, 6, 8, 10% (*w*/*w*) PPF	Reduced lipid content, increased phenolics and dietary fiber, decreased the glycemic index, and improved texture	[133]
	Chickpea flour-based snack	5% (*w*/*w*) PP powder	Enhanced nutritional profile by increasing dietary fiber, minerals, phenolic compounds, and protein content, while also improving oxidative stability and maintaining product quality during storage.	[134]
**Vegetable oils**	Soybean oil	1.25% (*v*/*v*) PP extract	Reduced peroxide, totox, and p-anisidine indices, as well as minimized oxidation products, enhancing oxidative stability of soybean oil	[50]
	Soybean oil	-	Lowered peroxide values, inhibited hexanal production, and improved oxidative stability of soybean oil across all tested chlorogenic acid concentrations	[135]
	Sunflower and soybean oil	5, 10, 50, 100, 200 ppm PP extract	Superior antioxidant activity over synthetic antioxidants, reducing peroxide and *p*-anisidine values in sunflower and soybean oils under accelerated oxidation	[136]
	Sunflower oil	0.32% (*w*/*v*) PP extract	Reduced FFA, PV, and p-anisidine values, minimizing oxidation products and enhancing the oxidative stability of sunflower oil, comparable to synthetic antioxidants	[137]
	Rapseed and sunflower oil	-	Reduced oxidation of rapeseed and sunflower oils, minimizing conjugated diene and volatile formation, and enhancing oxidative stability	[138]
**Meat and fish products**	Irradiated lamb meat	0.04% (*w*/*w*) PP extract	Increased total phenolic and chlorogenic acid content, enhanced radical scavenging activity and oxidative stability, and reduced lipid peroxidation in irradiated lamb meat	[139]
	Beef meatballs	5, 10, 15, 20% (*w*/*w*) PP powder	Increased crude dietary fiber, protein, and ash content, enhanced water-holding capacity and cooking yield, and reduced cooking loss and energy intake in beef meatballs	[140]
	Hamburger	0.75, 1.5, 2.25% (*w*/*w*) PPF	Enhanced nutritional profile and technological properties of bovine hamburgers, with minimal impact on sensory attributes and overall acceptance	[141]
	Pork patties	2, 5, 10% (*w*/*w*) PP powder	Increased antioxidant activity and total phenolic content, enhanced water-holding capacity and oxidative stability, and reduced pH changes, lipid oxidation, and cooking loss in pork patties	[142]
	Minced horse mackerel	0.24, 0.48% (*w*/*w*) PP extract	Reduced lipid and protein oxidation, increased phenolic content, and preserved α-tocopherol and amino acid residues	[87]
	Cooked salmon	0.02% (*w*/*w*) **	Reduced lipid oxidation in cooked salmon, with Russet PP extract showing the highest TBARS inhibition (83.4%), outperforming synthetic antioxidants and enhancing oxidative stability during storage	[75]
**Dairy products**	Yogurt	0.8% (*w*/*w*) PP powder	Improved nutritional, antioxidant, and sensory qualities, with increased phenolic content and antioxidant capacity in yogurt	[82]
	Cow butter	0.3% (*w*/*w*) PP extract	Improved oxidative stability, shelf-life, and microbial growth control in cow butter	[143]
	Processed cheese	1,2,4% (*w*/*w*) PP powder	Increased protein and ash content, enhanced antioxidant activity, and reduced meltability and oil separation, modifying texture and functional properties of processed cheese	[144]

(**): express as phenolic content; (-): not applicable; PP: potato peel; PPF: potato peel flour; FFA: free fatty acids; PV: peroxide value.

## Data Availability

No new data were created or analyzed in this study. Data sharing is not applicable to this article.

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
