# Peer review of "The Valorization of Potato Peels as a Functional Ingredient in the Food Industry: A Comprehensive Review"

_foods, 2025, doi:10.3390/foods14081333_

Round 1

Reviewer 1 Report

Comments and Suggestions for Authors

The submitted review covers the functional applications of the potato peel in the food industry by examining its chemical profile, extraction methods, and biological activities. The authors have thoroughly covered the composition, extraction, and applications of potato peel in the food industry. The review is very well organized, but some clarifications are needed.

Specific comments:

All abbreviations mentioned in the text should be defined; please check.

Page 3, at the end of the introduction section, the aim of the review should be clearly stated.

Page 13, line 350, Errore. L'origine riferimento non è stata trovata please, remove from the text.

Page 14, consider including a few lines about each non-conventional extraction methods from the economic and environmental point of view.

In section 4, discuss the influence of the variety of the potato on the biological activity.

Include in Table 8 , the % of the addition of potato peel to different food products, already mentioned.

line 701, avoid using literature references in the conclusion section.

Pages 18-35. Organize references to MDPI requirements.

Author Response

Dear Reviewer,

Thank you for your valuable feedback. We appreciate the time and effort you have dedicated to reviewing our manuscript.

  1. All abbreviations mentioned in the text should be defined; please check.

We have revised the manuscript according reviewer’s suggestion. All abbreviations have been defined.

  1. Page 3, at the end of the introduction section, the aim of the review should be clearly stated.

Thanks for your proposal. We have corrected the introduction as follows:
(lines 107-116): Considering the growing interest in research on agricultural by-products and their potential benefits, this review aims to comprehensively assess the current state of knowledge on the valorisation of potato peels (PP) as additive in functional food. Specifically, it will analyse their nutritional and bioactive composition, potential applications in the food industry, and their role in promoting sustainability through waste reduction and circular bioeconomy strategies. This review will also discuss the main challenges and future perspectives related to the utilization of PP-derived bioactive compounds in food product development. In light of global challenges such as food security, climate change, and resource efficiency, these findings highlight the importance of valorising agro-industrial by-products for both environmental and health-related benefits.

  1. Page 13, line 350, Errore. L'origine riferimento non è stata trovata please, remove from the text

Thank you for pointing this out. We have corrected the issue: at that specific point, the reference was labelled as Table 5 (now at line 355). Noticing this error also allowed us to identify and correct a similar mistake at line 148, where Table 1 had the same notation issue. Additionally, we realized that the section order between 4.3 and 4.4 was incorrect due to a duplicate 4.3. We have now fixed this issue as well in the revised manuscript (line 521).

  1. Page 14, consider including a few lines about each non-conventional extraction methods from the economic and environmental point of view

Thank you for your insightful suggestion. In the revised manuscript we have included and highlighted additional lines discussing the economic and environmental aspects of each non-conventional extraction method. (Section 3.2, line 395-398, line 413-415,  line 429-431, line 446-450)

  1. In section 4, discuss the influence of the variety of the potato on the biological activity.

We have revised the manuscript as follows (line 485-490): “Similarly, a study by Ansari et al. investigated the antioxidant potential of the Lady Rosetta (LR) variety, characterized by red skin, and the FT-1533 variety, with brown skin [106]. Their findings highlighted that LR exhibited higher antioxidant activity, attributed to its greater phenolic compound content. These results align with previous studies such as Franková, where the red-skinned Cecil variety displayed among the highest phenolic contents, along with the purple-skinned Violet Queen variety [62].”

Additionally, we have incorporated data on the antibacterial properties (section 4.3) of the Fianna variety (line 533) as reported by Silva-Beltrán et al. However, for other biological activities, no significant differences among potato varieties have been reported in the literature or identified in our review.

  1. Include in Table 8, the % of the addition of potato peel to different food products, already mentioned.

Thank you for your valuable suggestion. In the revised manuscript we have added a column in Table 8 indicating the percentage of potato peel used in the different food products.

  1. line 701, avoid using literature references in the conclusion section.

All the references were removed from text (now at line 728)

  1. Pages 18-35. Organize references to MDPI requirements.

We sincerely apologize for the problem. We initially believed we had used the correct citation style. However, after reviewing the guidelines, we have now accessed the correct reference format through the page "https://www.mdpi.com/authors/references" and implemented the URL of the updated style in Mendeley.

Please, find attached response to your questions and suggestions.

Reviewer 2 Report

Comments and Suggestions for Authors

I send a review of manuscript (ID- foods-3564640) of the authors: Domizia Vescovo, Cesare Manetti, Roberto Ruggieri, U. Gianfranco Spizzirri, Francesca Aiello, Maria Martuscelli, Donatella Restuccia „Valorisation of Potato Peels as Functional Ingredient in Food Industry: A Comprehensive Review”.

I think the manuscript covers a very interesting area of scientific research. The above study is an interesting compilation of information on current research topics related to the issue of sustainable waste reduction strategy and improving the efficiency of natural resource use. It deals with important issues related to the management of waste from the food industry, including potato peelings, which are rich in various compounds. It also discusses the possibilities of recovering these compounds in an ecological way. The authors should make a minor revision.

  1. Introduction

Page 2, lines 88-89 - Potatoes can grow in all types of environments…- I suggest the change on to - I suggest the changing on to - Potatoes can grow in different climatic and growing conditions and …

Page 3, line 107 - … PP as ingredients for food functionalization  - I suggest the change on to - …PP    as additives in functional food.

Page 6, lines 207 - I think the authors should point out that the glycoalkaloids (α-solanine and α-chaconine) contained in potato tubers, and especially in the peel, are dangerous to humans. They become toxic when their total amount reaches about 200 mg-kg-1 fresh weight FW, causing, for example: neurological disorders such as apathy, drowsiness and disorientation, and can even be fatal.

Page 7, line 255 - …pigmented potatoes…- I suggest to use – …coloured-fleshed potatoes…

End of page number 7 - Change in the numbering of the pages (incorrect numbering in the next part of the manuscript).

Line 354 - Please include a reference to Table 5 in the manuscript text.

Lines 379 – 417; Please increase font size or do not use subscript, for unit - ...mg GAE g-1...

Line 512 , line 587; line 590; ect. - … Friedman et al. tested...; Jacinto et al….; Ben Jeddou et al. - please add the brackets and number of this literature. Please check it throughout the manuscript.

References

All references are cited in the text. However, please prepare a list of publications according to the requirements. Please read the guidelines carefully to see how it should be written.

Author Response

Dear Reviewer,

Thank you for taking the time to review our manuscript and for your valuable comments.

Please find attached our responses to your questions and suggestions.
